# The tower of Kontsevich deformations for Nambu-Poisson structures on $\mathbb{R}^d$: Dimension-specific micro-graph calculus

Ricardo Buring[1°] and Arthemy V. Kiselev[2⋆§]

**1** Institut für Mathematik, Johannes Gutenberg–Universität,
Staudingerweg 9, D-55128 Mainz, Germany
**2** Bernoulli Institute for Mathematics, Computer Science and Artificial Intelligence,
University of Groningen, P.O. Box 407, 9700 AK Groningen, The Netherlands

⋆ A.V.Kiselev@rug.nl

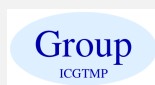

## Abstract

In Kontsevich's graph calculus, internal vertices of directed graphs are inhabited by multi-vectors, e.g., Poisson bi-vectors; the Nambu-determinant Poisson brackets are differential-polynomial in the Casimir(s) and density $\varrho$ times Levi-Civita symbol. We resolve the old vertices into subgraphs such that every new internal vertex contains one Casimir or one Levi-Civita symbol $\times\, \varrho$. Using this micro-graph calculus, we show that Kontsevich's tetrahedral $\gamma_3$-flow on the space of Nambu-determinant Poisson brackets over $\mathbb{R}^3$ is a Poisson coboundary: we realize the trivializing vector field $\vec{X}$ over $\mathbb{R}^3$ using micro-graphs. This $\vec{X}$ projects to the known trivializing vector field for the $\gamma_3$-flow over $\mathbb{R}^2$.

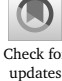

## Contents

---

° **Address for correspondence**: Team MATHEXP, Centre INRIA de Saclay Île-de-France, Bât. Alan Turing, 1 rue Honoré d'Estienne d'Orves, F-91120 Palaiseau, France
§ **Present address**: Institut des Hautes Études Scientifiques (IHÉS), 35 route de Chartres, Bures-sur-Yvette, F-91440 France

# 1 Introduction

Kontsevich introduced [5] a universal – for any affine Poisson manifold of dimension $d$ – construction of infinitesimal symmetries for the Jacobi identity: for suitable cocycles $\gamma$ in the graph complex, one obtains bi-vector flows $\dot{P} = Q_\gamma([P])$ with differential-polynomial right-hand sides (with respect to components $P^{ij}(\boldsymbol{x})$ of Poisson structures $P \in \Gamma(\bigwedge^2 TM_{\text{aff}}^d)$). We detect in [4] that for the tetrahedral graph cocycle $\gamma_3$ from [5] and for the pentagon-wheel graph cocycle $\gamma_5$ (see [3]), the corresponding flows (see [1,2]) have a well-defined restriction to the subclass of Nambu-determinant Poisson brackets[1] $P(\varrho,[\boldsymbol{a}])$ on $\mathbb{R}^d$ at least in the following three cases: (i) $\gamma_3$-cocycle flow $\dot{P} = Q_\gamma([P])$ for $P(\varrho,[\boldsymbol{a}])$ over $\mathbb{R}^3$, (ii) the same $\gamma_3$-cocycle and the flow of $P(\varrho,[\boldsymbol{a}_1],[\boldsymbol{a}_2])$ over $\mathbb{R}^4$, and (iii) the next, $\gamma_5$-cocycle flow for $P(\varrho,[\boldsymbol{a}])$ over $\mathbb{R}^3$.

To study the (non)triviality of Kontsevich's graph flows in the second Poisson cohomology group for Nambu–determinant brackets $\{\cdot,\cdot\}_{P(\varrho,[\boldsymbol{a}])}$, we consider the coboundary equation,

$$Q_\gamma([P])([\varrho],[\boldsymbol{a}]) = [\![P(\varrho,[\boldsymbol{a}]),\vec{X}([\varrho],[\boldsymbol{a}])]\!], \tag{1}$$

upon vector fields $\vec{X}([\varrho],[\boldsymbol{a}])$ with differential-polynomial coefficients over $\mathbb{R}^d$. We discovered in [4] that the $\gamma_3$-flow over $\mathbb{R}^3$ is trivial w.r.t. a unique solution $\vec{X}$ mod $[\![P,H([\varrho],[\boldsymbol{a}])]\!]$. In this text we explain how, for $\gamma = \gamma_3$ and $d = 2$, a solution of (1) is constructed using Kontsevich's graphs, and then, for $d = 3$, how the trivializing vector field $X^{\gamma_3}$ is found by using *micro-graphs* that resolve $\varrho(\boldsymbol{x}) \cdot$ Levi-Civita symbol against the Casimir(s) $a_\ell$ within copies of Nambu-determinant Poisson brackets $\{\cdot,\cdot\}_{P(\varrho,[\boldsymbol{a}])} = \varrho(\boldsymbol{x}) \cdot \sum_{i_1,\ldots,i_d=1}^d \varepsilon^{\vec{i}} \cdot \partial_{i_1}(a_1) \cdot \ldots \cdot \partial_{i_{d-2}}(a_{d-2}) \cdot \partial_{i_{d-1}} \otimes \partial_{i_d}$ in the vertices of Kontsevich's directed graphs for the flow $Q_\gamma([P])$.

# 2 Preliminaries: The tetrahedral flow $\dot{P} = Q_{\gamma_3}(P)$ over twofolds

Kontsevich's directed graphs are built of $n \geqslant 0$ wedges $\xleftarrow{L} \bullet \xrightarrow{R}$, usually drawn in the upper half-plane $\mathbb{H}^2$, over $m \geqslant 0$ ordered sinks along $\mathbb{R} = \partial \mathbb{H}^2$; tadpoles are allowed. Leibniz graphs are akin: the out-degrees of all but one (or more) vertices equal 2 yet there is (at least) one aerial vertex of out-degree 3 and its outgoing edges are ordered Left $\prec$ Middle $\prec$ Right.[2]

We shall study only those flows $\dot{P} = Q([P])$ on spaces of bi-vectors $P \in \Gamma(\bigwedge^2 TM_{\text{aff}}^{d<\infty})$ which are encoded by Kontsevich's graphs. From [5] (cf. [2]) we know that from suitable *cocycles* $\gamma$ in the Kontsevich graph complex, one obtains the flows[3] $\dot{P} = Q_\gamma([P])$ which preserve the (sub)set of *Poisson* bi-vectors on $M_{\text{aff}}^d$. The tetrahedron $\gamma_3$ and pentagon-wheel cocycle $\gamma_5$, see [2], are examples of graph cocycles giving such flows.

---

[1] The Nambu-determinant Poisson brackets (with $\varrho \not\equiv 1$ and Casimir(s) $a_\ell$) of $f, g \in C^1(\mathbb{R}^d)$ are, e.g.,

$$\{f,g\}_{P(\varrho,[a])} = \varrho(x,y,z) \cdot \begin{vmatrix} a_x & f_x & g_x \\ a_y & f_y & g_y \\ a_z & f_z & g_z \end{vmatrix}, \quad \text{on } \mathbb{R}^3 \ni \boldsymbol{x} = (x,y,z),$$

likewise $\{f,g\}_{P(\varrho,[a_1],[a_2])} = \varrho(x^1,x^2,x^3,x^4) \cdot \det(\partial(a_1,a_2,f,g)/\partial(x^1,x^2,x^3,x^4))$ on $\mathbb{R}^4$, and so on; all such formulas are tensorial w.r.t. coordinate transformations (as $\varrho(\boldsymbol{x}) \cdot \partial_{x^1} \wedge \ldots \wedge \partial_{x^d}$ is a top-degree multivector on $\mathbb{R}^d$).

[2] For example, the tripod is a Leibniz graph; like every Leibniz graph, it expands to a linear combination of Kontsevich graphs, namely to the Jacobiator $\frac{1}{2}[\![P,P]\!]$ for a bi-vector $P$ whose copies are realized by wedges ([1]).

[3] The formula of Kontsevich's graph flow $\dot{P} = Q_\gamma([P])$ can depend on a choice of representative $\gamma$ for the graph cohomology class $[\gamma]$. Fortunately, the vertex-edge bi-gradings (4,6) for $\gamma_3$ and (6,10) for two graphs in $\gamma_5$ are not yet big enough to provide room for any nonzero coboundaries (from nonzero graphs on 3 vertices and 5 edges or on 5 vertices and 9 edges, respectively). In other words, the known markers for $[\gamma_3]$ and $[\gamma_5]$ are in fact uniquely defined up to a nonzero multiplicative constant; we prove this by listing all the admissible (non)zero "potentials" and by taking their vertex-expanding differentials in the graph complex. This is why, in our present study of the $\gamma_3$- and $\gamma_5$-flows on the spaces of Nambu–Poisson brackets, we do not care about a would-be response of trivializing vector fields $X^\gamma$ in (1) to shifts of the marker cocycle $\gamma$ within its graph cohomology class $[\gamma]$.

**Example 1** ( [5] and [1]). The tetrahedral-graph flow $\dot{P} = Q_\gamma(P)$ on the space of Poisson bi-vectors $P$ over any $d$-dimensional affine Poisson manifold $M_{\text{aff}}^{d \geqslant 2}$ is encoded by the linear combination of three directed graphs,[4]

$$Q_{\gamma_3} = 1 \cdot \big(0,1;2,4;2,5;2,3\big) - 3 \cdot \big(0,3;1,4;2,5;2,3 + 0,3;4,5;1,2;2,4\big), \qquad (2)$$

with a copy of the Poisson bi-vector in each internal vertex 2,3,4,5 from which two decorated arrows, Left $\prec$ Right matching the bi-vector indices, are issued to the designated arrowhead vertices. Sink vertices 0 and 1 contain the arguments of bi-vector $Q_{\gamma_3}(P)$; vertices 2,3,4,5 of the tetrahedron $\gamma_3$ itself are internal.

The graph construction of the $\gamma_3$-flow $\dot{P} = Q_{\gamma_3}(P)$ works in any dimension of the affine Poisson manifold $M_{\text{aff}}^d$ at hand. Now if $d = 2$, this infinitesimal deformation of bi-vectors on twofolds is known to be trivial in the second Poisson cohomology, thus amounting to an infinitesimal change of local coordinates (performed along the integral trajectories of the trivializing vector field $\vec{X}$ on $M_{\text{aff}}^2$).

**Proposition 1** ( [1, App. F]). **1.** In dimension $d = 2$ (where every bi-vector $P = \varrho(x,y) \cdot (\partial_x \otimes \partial_y - \partial_y \otimes \partial_x)$ is Poisson,[5] in absence of nonzero Jacobiator tri-vectors), Kontsevich's tetrahedral flow $\dot{P} = Q_{\gamma_3}(P)$, encoded by three graphs in (2), is Poisson-trivial,

$$Q_{\gamma_3}\big(P(\varrho)\big) - [\![P(\varrho), \vec{X}]\!] = 0 \in \Gamma\big(\textstyle\bigwedge^2 T M_{\text{aff}}^2\big), \qquad \vec{X} = X^{\gamma_3}(P), \quad X^{\gamma_3} = \text{⊘⃗}, \qquad (3)$$

with respect to the class of 1-vector $\vec{X} = \partial_j\big(\partial_k \partial_m(P^{ij}) \cdot \partial_n(P^{k\ell}) \cdot \partial_\ell(P^{mn})\big) \partial_i$ (modulo Hamiltonian vector fields $[\![P(\varrho), H([\varrho])]\!]$), encoded by the "sunflower" linear combination[6] of Kontsevich graphs $X^{\gamma_3} = \sum_{a=1}^{3} \big(0,a;1,3;1,2\big) = \big(0,1;1,3;1,2\big) + \big(0,2;1,3;1,2\big) + \big(0,3;1,3;1,2\big) = \big(0,1;1,3;1,2\big) + 2 \cdot \big(0,2;1,3;1,2\big)$.

**2.** In dimension $d = 2$, Poisson-coboundary equation (3) is valid as an equality of bi-vectors with differential-polynomial coefficients (w.r.t. $\varrho$ in bi-vector $P$), but its left-hand side cannot be expressed as a linear combination of zero Kontsevich graphs and Leibniz graphs (that is, differential consequences of the Jacobi identity, see [1]).

*Proof.* Equality (3) in Part 1 of Proposition 1 is verified in $d = 2$ by straightforward calculation with differential-polynomial coefficient (multi)vectors.

To explore *why* equality (3) is valid in $d = 2$, let us inspect whether it is the standard, working in all dimensions $d \geqslant 2$, Leibniz-graph mechanism that would ensure the vanishing, $Q_{\gamma_3}\big(P(\varrho)\big) - [\![P(\varrho), \vec{X}([\varrho])]\!] \doteq 0$ for any $\varrho(x,y)$, by force of the Jacobi identity realized by Kontsevich graphs. (We claim that it is *not only* this mechanism which does the job.)

For this, we list all connected directed graphs built over two sinks of in-degree 1 from one trident and two wedges, without co-directed double or triple edges, and with none or one tadpole. (The Jacobiator vertex with three outgoing edges will be expanded to the linear

---

[4]The encoding of each Kontsevich directed graph is an ordered list of ordered pairs of target vertices for the edges issued from the ordered set of arrowtails (here, $\{2,3,4,5\}$), that is of aerial vertices.

[5]For the same reason, the Poisson condition – trivial in $d = 2$ – is preserved by *any* Kontsevich bi-vector graph (not necessarily obtained from a cocycle, on $n$ vertices and $2n-2$ edges, w. r. t. the graph differential [3,5]). Yet, Proposition 1 condensed to Eq. (3) is not a tautology since the second Lichnerowicz–Poisson cohomology does not vanish *a priori* over $d = 2$; see, e. g., Ph. Monnier *Poisson cohomology in dimension two*, Israel J. Math. **129** (2002) 189–207 (Preprint `arXiv:math.DG/0005261`).

[6] The 'sunflower' 1-vector graph in Eq. (3) expands, by the Leibniz rule, to the linear combination of two Kontsevich graphs, one of them with a 1-cycle (or *tadpole*). This nonzero graph with tadpole in $X^{\gamma_3}$ survives in the bracket $[\![P, X^{\gamma_3}]\!]$, yet no tadpoles are present in the three graphs of the $\gamma_3$-flow $Q_{\gamma_3}$. The disappearance of the tadpole from $Q_{\gamma_3}\big(P(\varrho)\big) - [\![P(\varrho), \vec{X}]\!]$ is due to a mechanism which will be explored.

combination of three Kontsevich's graphs, see [1].) There are three admissible Leibniz graphs without tadpoles,

$$1. \quad (3,4;2,4;0,1,2), \qquad 2. \quad (1,3;2,4;0,2,3), \qquad 3. \quad (1,4;2,4;0,2,3),$$

where $0, 1$ are the sinks, vertices $2, 3$ are wedge tops, and vertex $4$ is the top of the trident. Likewise, we have nine admissible Leibniz graphs with one tadpole and sinks of in-degree 1:

$$
\begin{array}{lll}
4. \quad (2,4;2,4;0,1,2), & 5. \quad (2,4;2,4;0,1,3), & 6. \quad (2,3;2,4;0,1,2), \\
7. \quad (2,3;2,4;0,1,3), & 8. \quad (1,2;2,4;0,2,3), & 9. \quad (1,3;2,3;0,2,3), \\
10. \quad (1,4;2,3;0,2,3), & 11. \quad (1,3;3,4;0,2,3), & 12. \quad (1,4;3,4;0,2,3)
\end{array}
$$

(There remain four connected Leibniz graphs with *two* tadpoles but they are irrelevant for our present attempt to balance Eq. (3) using the topologies of Kontsevich graph expansions in the right-hand side).

When the wedge graph $P$ acts on the 'sunflower' 1-vector graph $X^{\gamma_3}$, three graph topologies with one tadpole are produced, among others (the tadpole is absent from all the other topologies that appear in $[\![P,X^{\gamma_3}]\!]$):

$$\text{A.} \quad (0,4;1,3;3,5;3,4), \qquad \text{B.} \quad (0,3;1,3;3,5;3,4), \qquad \text{C.} \quad (2,5;2,4;2,3;0,1). \qquad (4)$$

In Kontsevich's nonzero graph B, vertex 3 has in-degree four (tadpole counted). But no expansion – of Leibniz graph from the above list of twelve – into Kontsevich graphs results in a graph with vertex of in-degree four. Independently, nonzero bi-vector graph C – with tadpole on vertex 2 at distance two from either sink and with double edge $3 \rightleftarrows 4$ – is not obtained in the Kontsevich graph expansion of any of the relevant bi-vector Leibniz graphs 4–7 from the above list. Thirdly, only the Leibniz graph 8 reproduces nonzero graph A, but the Kontsevich graph expansion of 8 contains five more graphs (with a tadpole at distance one from the sink), none of which appears in the linear combination $Q_{\gamma_3} - [\![P,X^{\gamma_3}]\!]$ of Kontsevich graphs. Therefore, we have that $Q_{\gamma_3} - [\![P,X^{\gamma_3}]\!] \neq \Diamond\big(P,P,\frac{1}{2}[\![P,P]\!]\big)$, for any values of coefficients $\in \mathbb{R}$ near Leibniz graphs in the right-hand side. The proof is complete. $\qquad\square$

The fact of Poisson trivialization of the tetrahedral-graph flow $\dot{P} = Q_{\gamma_3}(P)$ in dimension two, $P \in \Gamma\big(\bigwedge^2 T M_{\mathrm{aff}}^2\big)$, impossible in this or any higher dimension $d \geqslant 2$ through the mechanism of vanishing by force of the Jacobi identity as the *only* obstruction, implies the existence of other analytic mechanism(s); those can work in combination with the former.

*Remark* 1. When all the sums over repeated indices, each running from 1 to the finite dimension $d = 2$, are expanded in the left-hand side of the identity $Q_{\gamma_3}\big(P(\varrho)\big) - [\![P(\varrho),\vec{X}\big(P(\varrho)\big)]\!] = 0 \in \Gamma\big(\bigwedge^2 T M_{\mathrm{aff}}^2\big)$, the arithmetic cancellation mechanism works several times: the l.-h.s. splits into sums which cancel out separately from each other. For instance, when the Poisson differential $[\![P(\varrho),\cdot]\!]$ acts on $\vec{X}$ and this 1-vector gets into a sink of the wedge graph of bi-vector, thus forming graphs A and B in particular (see Eq. (4)), the wedge top coefficient $\varrho$ has no derivative(s) falling on it. Yet no such terms occur at $d = 2$ in the graphs of $Q_{\gamma_3}$ with strictly positive in-degrees of internal vertices. Hence the two parts – with(out) zero-order derivative of $\varrho$ – of the differential-polynomial coefficient in the identity's l.-h.s. vanish simultaneously.

*Remark* 2 (on $GL(\infty)$-invariants parent to $GL(d)$-invariants). Kontsevich's construction of tensors from Poisson bi-vectors by using graphs is well-behaved under affine coordinate changes on the underlying manifold, thanks to contraction of upper indices of Poisson tensors $(P^{ij})$ in the arrowtail vertices and of lower indices in derivations at the edge arrowheads. The Jacobians of coordinate changes then belong to the general linear group, while the affine shifts are not felt at all by the index contractions. But, uniform over the dimensions $d$ of affine

Poisson manifolds, the graph technique builds invariants of linear representations for $GL(\infty)$. Linearly independent as graph expressions, such invariants can be linearly tied after projection to a given finite dimension $d$, where they become tensor-valued $GL(d)$-invariants.[7]

To conclude this section, we note that the original graph language of [5] for construction of tensor-valued invariants of $GL(\infty)$ is no longer enough for the study of Poisson (non)triviality for graph-cocycle flows on spaces of Poisson brackets over affine manifolds of given dimension $d$. To manage the bound $d < \infty$, one must either take a quotient over the (unknown) new linear relations between Kontsevich's graphs, or work *ab initio* with $GL(d)$-invariants.

We note also that in a given dimension, the problem of Poisson (non)triviality for universal flows (from [5] and subsequent work [2]) itself is meaningful: *a priori* nontrivial deformation can become trivial for a class of Poisson geometries at given $d$. Let us verify the triviality of the tetrahedral-graph flow $\dot{P}([\varrho],[a]) = Q_{\gamma_3}\big(P(\varrho,[a])\big)$ on the space of Nambu-determinant Poisson bi-vectors $P(\varrho,[a])$ over an affine threefold $\mathbb{R}^3$.

# 3 Nambu-determinant Poisson brackets: The $\gamma_3$-flow over $\mathbb{R}^3$

**Definition 1.** The *Nambu-determinant Poisson bracket* on $\mathbb{R}^{d \geqslant 3}$ is the derived bi-vector $P(\varrho,[a])$ $\overset{\text{def}}{=} [[[\ldots[[\varrho \cdot \partial_{x^1} \wedge \ldots \wedge \partial_{x^d}, a_1]], \ldots]], a_{d-2}]]$, where $\varrho(\boldsymbol{x}) \cdot \partial_{\boldsymbol{x}}$ is a $d$-vector field and scalar functions $a_\ell$ are Casimirs ($1 \leqslant \ell \leqslant d-2$). In global (e.g., Cartesian) coordinates $x^1, \ldots, x^d$ on $\mathbb{R}^d$, the Nambu bracket of $f, g \in C^1(\mathbb{R}^d)$ is expressed by the formula

$$\{f, g\}_{P(\varrho,[a])} = \varrho(\boldsymbol{x}) \cdot \sum_{i_1 \ldots, i_d = 1}^{d} \varepsilon^{\vec{\imath}} \cdot \partial_{i_1}(a_1) \cdots \partial_{i_{d-2}}(a_{d-2}) \cdot \partial_{i_{d-1}}(f) \cdot \partial_{i_d}(g), \qquad (5)$$

where $\varepsilon^{\vec{\imath}} = \varepsilon^{i_1, \ldots, i_d}$ is the Levi-Civita symbol on $\mathbb{R}^d$: $\varepsilon^{\sigma(1, \ldots, d)} = (-)^\sigma$ for $\sigma \in S_d$, else zero.[8]

*Remark* 3. Nambu–Poisson brackets on $\mathbb{R}^{d \geqslant 3}$ can be obtained from Nambu–Poisson brackets on $\mathbb{R}^{d+1}$ by taking $a_{d-1} = \pm x^{d+1}$ on $\mathbb{R}^{d+1}$ and by excluding the last Cartesian coordinate $x^{d+1}$ from the list of arguments for $\varrho(\boldsymbol{x})$ and $a_1, \ldots, a_d(\boldsymbol{x})$. • By doing the above for $d+1 = 3$, one obtains a generic bi-vector $P = \varrho(x^1, x^2) \partial_{x^1} \wedge \partial_{x^2}$, which is Poisson on $\mathbb{R}^2$, and the (Nambu) Poisson bracket $\{f, g\}(x, y) = \varrho(x, y) \cdot (f_x \cdot g_y - f_y \cdot g_x)$.

We recall from [4, §4.1] that, given a suitable graph cocycle $\gamma$ (e.g., $\gamma_3$ which we take here), Kontsevich's $\gamma$-flow $\dot{P} = \text{O}\vec{\text{r}}(\gamma)(P^{\otimes \#\text{Vert}(\gamma)})$ restricts to the set of Nambu–Poisson bi-vectors $P(\varrho,[a])$ such that the velocity of a Casimir $a_\ell$ is still encoded by Formality graphs [4, Proposition 2]: $\dot{a}_\ell = \text{O}\vec{\text{r}}(\gamma)(P \otimes \cdots \otimes P \otimes a_\ell)$, whence the velocity $\dot{\varrho}([\varrho],[a])$ is expressed from the known $\dot{a}$ and $\dot{P}$ (see [4, Corollary 3]). The Leibniz rule, balancing $\dot{P}$ with $\dot{\varrho}, \dot{a}$ for $P$ linear in $\varrho$ and the first jets of all $a_\ell$, is then a tautology.

Independently, if $\vec{Y}$ is any $C^1$-vector field on $\mathbb{R}^d$ with Nambu–Poisson bi-vectors $P(\varrho,[\boldsymbol{a}])$, then the evolution $L_{\vec{Y}}(a_\ell) = [[\vec{Y}, a_\ell]]$ of scalar functions and $L_{\vec{Y}}(\varrho \cdot \partial_{\boldsymbol{x}}) = [[\vec{Y}, \varrho \cdot \partial_{\boldsymbol{x}}]]$ of $d$-

---

[7]The main example is given by the linear independence (modulo zero graphs and Leibniz graphs) of three Kontsevich graphs in the $\gamma_3$-flow and, on the other hand, of the Poisson-exact bi-vector graphs $[[P, X^{\gamma_3}]]$ with the 'sunflower' 1-vector $X^{\gamma_3}$: an insoluble equation for graphs, $Q_{\gamma_3} - [[P, X^{\gamma_3}]] = 0$, turns at $d = 2$ into an identity of bi-vector's differential-polynomial coefficients, $Q_{\gamma_3}\big(P(\varrho)\big) - [[P(\varrho), \vec{X}\big(P(\varrho)\big)]] \equiv 0$, for the Poisson structure $P(\varrho) = \varrho(x, y) \cdot (\partial_x \otimes \partial_y - \partial_y \otimes \partial_x)$ in every chart of the affine twofold $M^2_{\text{aff}}$.

[8]The usual view of Nambu–Poisson bracket (5) on $\mathbb{R}^d$ is that the Jacobian determinant is multiplied by an arbitrary factor $\varrho(x^1, \ldots, x^d)$ which behaves appropriately under coordinate changes $\boldsymbol{x} \rightleftarrows \boldsymbol{x}'$. Our viewpoint is that $\varrho(\boldsymbol{x}) \partial_{x^1} \wedge \ldots \wedge \partial_{x^d}$ is a top-degree multi-vector on $\mathbb{R}^d$ for any $d \geqslant 2$, so that for all dimensions greater than two, the Nambu-determinant Poisson bi-vector is *derived*: $P(\varrho,[\boldsymbol{a}]) = [[\ldots[[\varrho \, \partial_{\boldsymbol{x}}, a_1]] \ldots, a_{d-2}]]$ with $d-2$ Casimirs $\boldsymbol{a}$. That is, Nambu structures generalize the bi-vector $\varrho(x, y) \cdot (\partial_x \otimes \partial_y - \partial_y \otimes \partial_x)$ from $\mathbb{R}^2$ to $\mathbb{R}^d$ for $d > 2$. The Casimirs $a_\ell$ Poisson-commute with any $f \in C^1(\mathbb{R}^d)$; the symplectic leaves are intersections of level sets $a_\ell = \text{const}(\ell) \in \mathbb{R}$, so that for any $d \geqslant 2$ these leaves are at most two-dimensional.

vectors correlates, by the Leibniz-rule shape of the Jacobi identity for the Schouten bracket $[[\cdot,\cdot]]$, with evolution $L_{\vec{Y}}(P) = [[\vec{Y}, P]]$ of Nambu bi-vector $P = [[\varrho \cdot \partial_x, \cdots a \cdots]]$, see [4, §2.1].

**Theorem 2** ( [4]). *In dimension $d = 3$, the tetrahedral-graph flow $\dot{P} = Q_{\gamma_3}(P)$ on the space of Poisson bi-vectors $P$ has a well-defined restriction to the subspace of Nambu-determinant Poisson bi-vectors $P(\varrho, [a])$ on $\mathbb{R}^3$, and this restriction is Poisson-cohomology trivial: $\dot{P}([\varrho], [a]) = [[P(\varrho, [a]), \vec{X}^{\gamma_3}([\varrho], [a])]]$. The equivalence class $\vec{X}^{\gamma_3}$ mod $[[P, H(\varrho, a)]]$ of trivializing vector field is represented by the vector $\vec{X} = \sum_{\vec{i}, \vec{j}, \vec{k}} \varepsilon^{\vec{i}} \varepsilon^{\vec{j}} \varepsilon^{\vec{k}} \cdot X_{\vec{i}\vec{j}\vec{k}}$ with*

$$
\begin{aligned}
X_{\vec{i}\vec{j}\vec{k}} = {} & 12\varrho\,\varrho_{x^{k_2}}\varrho_{x^{i_1}x^{j_1}}a_{x^{k_3}}a_{x^{i_2}x^{j_2}}a_{x^{i_3}x^{j_3}} \cdot \partial/\partial x^{k_1} + 48\varrho\,\varrho_{x^{j_3}}\varrho_{x^{i_1}x^{j_1}}a_{x^{k_3}}a_{x^{i_2}x^{j_2}}a_{x^{i_3}x^{k_1}} \cdot \partial/\partial x^{k_2} \\
& + 8\varrho_{x^{j_2}}\varrho_{x^{i_1}x^{k_1}}\varrho_{x^{i_2}x^{k_2}}a_{x^{i_3}}a_{x^{j_3}}a_{x^{k_3}} \cdot \partial/\partial x^{j_1} - 40\varrho_{x^{i_3}}\varrho_{x^{j_2}}\varrho_{x^{i_1}x^{k_1}}a_{x^{j_3}}a_{x^{k_3}}a_{x^{i_2}x^{k_2}} \cdot \partial/\partial x^{j_1} \\
& + 8\varrho_{x^{i_3}}\varrho_{x^{j_2}}\varrho_{x^{k_3}}a_{x^{j_3}}a_{x^{i_1}x^{k_1}}a_{x^{i_2}x^{k_2}} \cdot \partial/\partial x^{j_1} + 24\varrho_{x^{j_2}}\varrho_{x^{k_3}}\varrho_{x^{i_1}x^{k_1}}a_{x^{i_3}}a_{x^{j_3}}a_{x^{j_1}x^{k_2}} \cdot \partial/\partial x^{i_2} \\
& - 12\varrho^2_{x^{k_2}}a_{x^{i_1}x^{j_1}}a_{x^{i_2}x^{j_2}}a_{x^{i_3}x^{j_3}x^{k_3}} \cdot \partial/\partial x^{k_1} + 24\varrho\,\varrho_{x^{j_2}}\varrho_{x^{k_1}}a_{x^{k_2}}a_{x^{i_1}x^{j_1}}a_{x^{i_3}x^{j_3}x^{k_3}} \cdot \partial/\partial x^{i_2} \\
& - 36\varrho\,\varrho_{x^{i_2}}\varrho_{x^{j_2}}a_{x^{k_2}}a_{x^{i_1}x^{j_1}}a_{x^{i_3}x^{j_3}x^{k_3}} \cdot \partial/\partial x^{k_1} + 8\varrho_{x^{i_2}}\varrho_{x^{j_1}}\varrho_{x^{k_1}}a_{x^{j_2}}a_{x^{k_2}}a_{x^{i_3}x^{j_3}x^{k_3}} \cdot \partial/\partial x^{i_1} \\
& - 8\varrho_{x^{j_1}}\varrho_{x^{k_1}}\varrho_{x^{i_3}x^{j_3}x^{k_3}}a_{x^{i_2}}a_{x^{j_2}}a_{x^{k_2}} \cdot \partial/\partial x^{i_1} ,
\end{aligned}
$$

*where $\vec{i} = (i_1, i_2, i_3)$, $\vec{j} = (j_1, j_2, j_3)$, $\vec{k} = (k_1, k_2, k_3)$ and $\varepsilon^{pqr}$ is the Levi-Civita symbol on $\mathbb{R}^3$.*

Our next finding in [4, Theorem 8] is that for the graph cocycle $\gamma_3$ and $d = 3$, the action of vector field $\vec{X}$ (which trivializes the $\gamma_3$-flow $\dot{P} = Q_{\gamma_3}([P]) = [[P, \vec{X}]]$ of Nambu brackets $P$ on $\mathbb{R}^3$) upon $P(\varrho, [a])$ factors through the initially known – from $\gamma_3$ – velocities of $a$ and $\varrho$: having solved (1) for $\vec{X}$, we then verified that $\dot{a} = [[a, \vec{X}]]$ and $\dot{\varrho} \cdot \partial_x = [[\varrho \cdot \partial_x, \vec{X}]]$.

By using this factorization – i.e. the lifting of the sought vector field's action on the elements of $P(\varrho, [a])$ – the other way round, we create an economical scheme to inspect the *existence* of trivializing vector field $\vec{X}$ for larger problems (i.e. for bigger graph cocycles or higher dimension $d \geqslant 3$). When this shortcut works, so that $\vec{X}$ is found, it saves much effort. Otherwise, to establish the (non)existence of $\vec{X}$ one deals with a larger PDE, namely Eq. (1).

**Definition 2.** Fix the dimension $d \geqslant 2$. A *micro-graph* is a directed graph built over $m \geqslant 0$ sinks, over $n \geqslant 0$ aerial vertices with out-degree $d$ and ordering of outgoing edges, and over $n$ items of $(d-2)$-tuples of aerial vertices with in-degree $\geqslant 1$ and no outgoing edges. • The correspondence between micro-graphs and differential-polynomial expressions in $\varrho, a_1, \ldots, a_{d-2}$ and the content of sink(s) is defined in the same way as the mapping of Kontsevich's graphs to multi-differential operators on $C^\infty(M^d_{\text{aff}})$, see [4, §2.2] or [5]. • Same as for Kontsevich graphs, a micro-graph is *zero* if it admits a sign-reversing automorphism, i.e. a symmetry which acts by parity-odd permutation on the ordered set of edges. But now, in finite dimension $d$ of $M^d_{\text{aff}}$, a micro-graph is *vanishing* if the differential polynomial (in $\varrho$ and $a_1$, $\ldots, a_{d-2}$), obtained by expanding all the sums over indices that decorate the edges, vanishes identically.[9]

**Example 2.** Nambu–Poisson brackets $P(\varrho, [a_1], \ldots, [a_{d-2}])$ on $\mathbb{R}^d$ are realized using micrographs, namely by resolving $\varrho(\boldsymbol{x}) \cdot \varepsilon^{i_1, \ldots, i_d}$ in one vertex against $d-2$ vertices with the Casimirs $a_1, \ldots, a_{d-2}$. The out-degree of vertex with $\varrho(\boldsymbol{x}) \cdot \varepsilon^{\vec{i}}$ equals $d$; the in-degree of each vertex with a Casimir equals 1 and its out-degree is zero: the Casimir vertices are terminal (not to be confused with the two sinks, of in-degree 1, for the Poisson bracket arguments). The ordered $d$-tuple of edges is decorated with summation indices: for the Levi-Civita symbol $\varepsilon^{i_1, \ldots, i_d}$ in their common arrowtail vertex, the range is $1 \leqslant i_\ell \leqslant d$ for $1 \leqslant \ell \leqslant d$.

*Remark* 4. If the wedge tops contain Nambu–Poisson bi-vectors $P(\varrho, [a])$ on $\mathbb{R}^d$, every Kontsevich graph expands to a linear combination of micro-graphs: the arrow(s) originally in-coming

---

[9]There are nonzero but still vanishing micro-graphs.

to an aerial vertex with a copy of $P$, now work(s) by the Leibniz rule over the $d-1$ vertices, with $\varrho \cdot \varepsilon^{\vec{\imath}}$ and with $a_1, \ldots, a_{d-2}$, in the subgraphs $P(\varrho, [a_1], \ldots, [a_{d-2}])$ of the micro-graph.[10]

**Example 3.** In $d = 3$, the micro-graph expansion of $Q_{\gamma_3}(P)$ for $P(\varrho, [a])$ over $\mathbb{R}^3$ consists of directed graphs on 2 sinks for $f$ and $g$, on four terminal vertices – for copies of the Casimir $a$ – without outgoing arrows, and on four vertices for $\varrho \cdot \varepsilon^{ijk}$ with three ordered outgoing edges. In every micro-graph in bi-vector $Q_{\gamma_3}(P)$ there are 12 edges, with exactly one going towards $f$ and one to $g$ in the sinks. To have a solution $\vec{X}$ of the equation $Q_{\gamma_3}(P(\varrho, [a])) = [\![P, \vec{X}]\!]$ using micro-graphs that encode $\vec{X}([\varrho], [a])$, we thus need micro-graphs on one sink, three terminal vertices with $a$, and three trident vertices for $\varrho \cdot \varepsilon^{ijk}$. Of the nine edges in each micro-graph, exactly one goes to the sink, so that $\vec{X}$ is a 1-vector.

**Example 4.** In $d = 3$, two Kontsevich's graphs of the 'sunflower' 1-vector (which trivializes the restriction of tetrahedral-graph flow (2) to the space of bi-vectors on an affine twofold) expand to a linear combination of 42 one-vector micro-graphs over one sink, three trident vertices, three terminal vertices, and $3 \times 3 = 9$ edges (one into the sink). A tadpole is met in $10 = 3 + 3 + 4$ such micro-graphs, and the other $32 = 8 \times 4$ have none.

Let us illustrate how the shortcut scheme works. We now tune a 1-vector field $\vec{X}(\varrho, [a])$ for the flow $\dot{P} = Q_{\gamma_3}([P])$ of $P(\varrho, [a])$ over $\mathbb{R}^3$ such that $\dot{a} = [\![a, \vec{X}]\!]$ and $\dot{\varrho} \cdot \partial_x = [\![\varrho \cdot \partial_x, \vec{X}]\!]$, whence we verify that $\dot{P} = [\![P, \vec{X}([\varrho], [a])]\!] \in \operatorname{im} \partial_P$ for the Nambu-determinant class of Poisson brackets on $\mathbb{R}^3$.

We first generate all suitable unlabeled micro-graphs (i.e. without distinguishing which sinks are for Casimirs) without tadpoles and with exactly one tadpole. Next, by deciding on the run which of the four sinks is the argument of 1-vector, we produce 366 1-vector fields with differential-polynomial coefficients in $\varrho$ and $a$, encoded by micro-graphs. Some of the coefficients are identically zero when the sums over three triples of indices in Levi-Civita symbols are fully expanded; there remain 244 nonvanishing marker micro-graphs in the ansatz for the trivializing vector field $\vec{X}$. Now, we do not attempt to solve the big problem $Q_{\gamma_3}(P) = [\![P, \vec{X}]\!]$ directly with respect to the 244 coefficients of nonvanishing marker micro-graphs. Instead, let us find a vector field $\vec{X}$, realized by 1-vector micro-graphs $X^\gamma$, which reproduces the known velocities [4, Eq. (11)] of $\varrho$ and Casimir $a$, that is, we solve the equations $\dot{a} = -[\![\vec{X}, a]\!]$ and $\dot{\varrho} \, \partial_x \wedge \partial_y \wedge \partial_z = [\![\varrho \, \partial_x \wedge \partial_y \wedge \partial_z, \vec{X}]\!]$ with respect to the coefficients in the micro-graph ansatz for $X^\gamma$. To determine exactly the number of equations in either linear algebraic system we keep track of the number of differential monomials appearing when $\vec{X}$ acts on either $a$ or $\varrho$ as above, and we recall also the differential monomials which already appeared in $\dot{a}$ and $\dot{\varrho}$ in the left-hand sides, that is in [4, Eq. (11)]. In this way, we detect that the linear algebraic system for $\dot{a}$ contains 2961 equations and the system for $\dot{\varrho}$ contains 6679 equations. Each equation is a balance of the coefficient of one differential monomial. We now merge these two systems of linear algebraic equations upon the coefficients of micro-graphs in the ansatz for the trivializing vector field $\vec{X}$, and we find a solution. Only 11 coefficients are nonzero. The analytic formula of this vector field is reported in Theorem 2. The three equalities, namely $\dot{a} = -[\![\vec{X}, a]\!]$ and $\dot{\varrho} \, \partial_x \wedge \partial_y \wedge \partial_z = [\![\varrho \, \partial_x \wedge \partial_y \wedge \partial_z, \vec{X}]\!]$ implying $Q_{\gamma_3}(P) = [\![P, \vec{X}]\!]$, are verified immediately. Here is the encoding[11] of the weighted sum $X^\gamma$ of these 11 micro-graphs which

---

[10]But not every micro-graph is obtained from a Kontsevich graph by resolving the old aerial vertices into subgraphs. This is what makes interesting our graphical rephrasing of the Poisson (non)triviality problem for the restriction of Kontsevich graph flows to the class of Nambu-determinant Poisson brackets.

[11]Vertices are labelled and the sink is indicated; for each micro-graph, its directed edges are listed in due ordering: e.g., $(6,6)$ is the tadpole $6 \to 6$ and $(0,4), (0,5), (0,6)$ is a trident Left $\prec$ Middle $\prec$ Right.

realize the trivializing vector field $\vec{X}$ for the tetrahedral flow $Q_{\gamma_3}(P) = [\![P, \vec{X}]\!]$ on the space of Nambu–Poisson structures over $\mathbb{R}^3$:

```
 16 * [(0,4), (0,5), (0,6), (5,0), (5,1), (5,6), (6,0), (6,2), (6,3)]      (sink 2),
 24 * [(0,4), (0,5), (0,6), (5,0), (5,1), (5,6), (6,1), (6,2), (6,3)]      (sink 2),
 16 * [(0,4), (0,5), (0,6), (5,0), (5,1), (5,2), (6,1), (6,3), (6,5)]      (sink 2),
-16 * [(0,4), (0,5), (0,6), (5,0), (5,1), (5,2), (6,1), (6,3), (6,5)]      (sink 4),
 12 * [(0,4), (0,5), (0,6), (5,1), (5,2), (5,6), (6,1), (6,2), (6,3)]      (sink 3),
-12 * [(0,4), (0,5), (0,6), (5,1), (5,2), (5,6), (6,1), (6,2), (6,3)]      (sink 4),
 24 * [(4,0), (4,1), (4,6), (5,0), (5,1), (5,2), (6,0), (6,2), (6,3)]      (sink 3),
-24 * [(4,0), (4,1), (4,6), (5,0), (5,2), (5,4), (6,0), (6,1), (6,3)]      (sink 2),
  8 * [(4,0), (4,1), (4,5), (5,0), (5,2), (5,6), (6,0), (6,3), (6,4)]      (sink 1),
 -8 * [(4,0), (4,1), (4,5), (5,2), (5,3), (5,6), (6,0), (6,1), (6,4)]      (sink 2),
  8 * [(0,4), (0,5), (0,6), (5,0), (5,1), (5,6), (6,2), (6,3), (6,6)]      (sink 2).
```

*Remark* 5. At the level of micro-graphs, the solution $X^{\gamma_3}$ contains a tadpole, i.e. a 1-cycle, in the last graph. In terms of differential polynomials this means the presence of a deriative $\partial_{x^i}$ acting on the coefficient $\varrho(\boldsymbol{x})$ near the Levi-Civita symbol $\varepsilon^{ijk}$ containing the index $i$ of the base coordinate $x^i$ in that derivative; that is, the last term in the vector field $\vec{X}$ contains $\partial_{x^i}(\varrho(\boldsymbol{x})) \cdot \varepsilon^{ijk}$.

**Proposition 3.** Without tadpoles in the micro-graph ansatz $X^{\gamma_3}$, there is no solution $\vec{X}$ to the trivialization problem $Q_{\gamma_3}(P(\varrho, [a])) = [\![P, \vec{X}]\!]$ at the level of differential polynomials.[12]

*Remark* 6. If $d = 2$ and (Nambu–)Poisson brackets on $\mathbb{R}^2$ are $\{f, g\}(x, y) = \varrho \cdot (f_x \cdot g_y - f_y \cdot g_x)$ as in Remark 3, the only possible Kontsevich 'sunflower' graphs, built from $n = 3$ wedges over one sink (see (3)), tautologically expand to a nontrivial linear combination of nonzero micro-graphs on three aerial vertices with $\varrho(x, y) \cdot \varepsilon^{i^\alpha j^\alpha}$, $1 \leqslant \alpha \leqslant 3$. Independently, the linear combination $X^\gamma$ of micro-graphs that encode the trivializing vector field $\vec{X}([\varrho], [a])$ for Kontsevich's $\gamma_3$-flow for Nambu bi-vectors $P(\varrho, [a])$ on $\mathbb{R}^3 \ni (x, y, z)$, see Theorem 2, under the reduction $a := z$ and $\varrho = \varrho(x, y)$ becomes a well-defined vector field on the plane $\mathbb{R}^2 \subset \mathbb{R}^3$: the $z$-component of $\vec{X}([\varrho(x, y)], [z])$ vanishes. Let us compare the two vector fields on $\mathbb{R}^2$.

**Proposition 4.** The old 'sunflower' vector field which trivializes the tetrahedral $\gamma_3$-flow for all Poisson brackets on $\mathbb{R}^2$ *coincides* with the new vector field $\vec{X}([\varrho(x, y)], [a = z])$ from the trivialization of $\gamma_3$-flow for the Nambu brackets $P(\varrho, [a])$ on $\mathbb{R}^3$ (both viewed as 1-vector fields on $\mathbb{R}^2$ with differential coefficients in $[\varrho]$). • Yet the linear combination $X_3^{\gamma_3}$ of micro-graphs over $d = 3$ contains *not only* and *not all* the expansions of Kontsevich's graphs from the 'sunflower' 1-vector into micro-graphs over $d = 3$.

*Proof.* The micro-graph expansion of the 'sunflower' graph is not enough in $d = 3$ because, in particular, the before-last micro-graph in $X_3^{\gamma_3}$, namely $(-8) \cdot [(4, 0), (4, 1), (4, 5), (5, 2), (5, 3), (5, 6), (6, 0), (6, 1), (6, 4)]$ with trident vertices $4, 5, 6$, sink $2$, and terminal vertices $0, 1, 3$, does not originate from either graph in the 'sunflower' $X_2^{\gamma_3}$. Indeed, the above micro-graph contains a 3-cycle $4 \to 5 \to 6 \to 4$ but no edge from 6 to either 5 or its Casimir 3 in a would-be expansion of $P(\varrho, [a])$ with 5 in the trident top.[13] $\qquad\square$

Let us examine how, by which mechanism(s), Eq. (1) is verified for the tetrahedral cocycle $\gamma_3$ and the respective flow of Nambu brackets on $\mathbb{R}^3$.

---

[12]By setting to zero the coefficients of micro-graphs with one tadpole, that is by excluding all the differential polynomials in $\varrho$ and $a$ which stem from those micro-graphs with a tadpole, we detect that the linear algebraic system for the coefficients of micro-graphs without tadpoles has no solution at all.

[13]Not all of the micro-graphs appearing in the expansion of Kontsevich's graphs $X_{d=2}^{\gamma_3}$ mod $[\![P, H]\!]$ are needed for a solution $X_3^{\gamma_3}$, see Example 4 and the eleven micro-graphs on p. 8.

**Lemma 5.** If $d = 3$ and $P(\varrho, [a])$ is Nambu, the Jacobiator tri-vector graph $\mathrm{Jac}(P)$ remains a nontrivial linear combination, $\mathrm{Jac}(P)([\varrho], [a])$, of 3 or 6 nonvanishing micro-graphs, each on $m = 3$ sinks of in-degree 1, on two trident vertices with $\varrho \cdot \varepsilon^{i_1^\alpha i_2^\alpha i_3^\alpha}$, and on two terminal vertices of in-degrees 1 and 2 (if $\varrho \equiv \mathrm{const}$) or $(1,1)$ and $(1,2)$ if $\varrho \not\equiv \mathrm{const}$.

**Proposition 6.** The trivialization mechanism, $Q_{\gamma_3}(P(\varrho, [a])) - [[P(\varrho, [a]), \vec{X}_3^{\gamma_3}([\varrho], [a])]] \doteq 0$, for the tetrahedral-graph flow on the space of Nambu-determinant Poisson bi-vectors $P(\varrho, [a])$ over $\mathbb{R}^3$ and for the linear combination of micro-graphs $X_3^{\gamma_3}$, does not amount *only to* the Leibniz (micro)graphs and zero (micro)graphs in the right-hand side of coboundary equation (1).

*Proof.* Suppose for contradiction that a linear combination $\Diamond$ of Leibniz and zero micro-graphs turns the coboundary equation, $Q_{\gamma_3} - [[P, X^{\gamma_3}]] = \Diamond([\varrho], [a], \frac{1}{2}[[P, P]])$, into an equality of bi-vector micro-graphs. By setting the Casimir $a := z$ we fix the values of indices decorating the arrows which run into the terminal vertices $a$; now for $\varrho(x, y)$ independent of $z$, the rest of each micro-graph becomes a Kontsevich graph over $d = 2$. Zero micro-graphs from dimension 3 remain zero over dimension 2. This dimensional reduction would yield a Leibniz-graph realization of the corresponding r.-h.s. for the two-dimensional problem from Part 2 of Proposition 1. But this is impossible; therefore, at least one new mechanism of differential polynomials' vanishing is at work. $\qquad\square$

## Conclusion

Kontsevich's symmetries $\dot{P} = Q_\gamma(P)$ of the Jacobi identity $\frac{1}{2}[[P, P]] = 0$ are produced from suitable graph cocycles $\gamma$ as described in [2, 5]. We study the (non)triviality of these flows w.r.t. the Poisson differential $\partial_P = [[P, \cdot]]$. The original formalism of [5] yields tensorial $GL(\infty)$-invariants; but in finite dimension $d$ of Poisson manifolds, these tensors, encoded by Kontsevich's graphs, can become linearly dependent, whence the 'incidental' trivializations (e.g., in $d = 2$). The graph language can be adapted to the subclass of Nambu-determinant Poisson brackets $P(\varrho, [a])$; Kontsevich's graph-cocycle flows do restrict to the Nambu subclass (see [4]). The new calculus of micro-graphs is good for encoding known flows and vector fields. Still the core task of this research is *finding* these fields $\vec{X}([\varrho], [a])$ or proving their non-existence over $\mathbb{R}^d$.

The calculus of micro-graphs is (almost) well-behaved[14] under the dimensional reduction $d \mapsto d - 1$ by the loss of one Casimir $a_{d-2} := x^d$ and last coordinate $x^d$ in $\varrho$ and all other Casimirs $a_\ell$, $\ell < d - 2$. The forward move, $d \mapsto d + 1$, is not well defined. *A priori* there is no guarantee that any solution $X_{d+1}^\gamma$ exists at all: the dimensions $d_0 < d_0 + 1$ can mark the threshold where the $GL(\infty)$-invariants from Kontsevich's graphs lose their linear dependence in lower dimensions $d \leqslant d_0$, and the flow $\dot{P}([\varrho], [a_1], \ldots, [a_{d-1}]) = Q_{d+1}^\gamma(P(\varrho, [a]))$ becomes Poisson-nontrivial is dimension $d_0 + 1$ and onwards.

**Lemma 7.** Suppose there exists a trivializing vector field $X_{d+1}^\gamma$ for a $\gamma$-flow of $P(\varrho, [a])$ over $\mathbb{R}^{d+1}$, and this solution projects – when $a_1 := x^3$, ..., $a_{d-1} := x^{d+1}$ – onto the known linear combination $X_{d=2}^\gamma$ of Kontsevich's graphs. Then $X_{d+1}^\gamma$ contains (at least) those micro-graphs from the expansion of $X_{d=2}^\gamma$ over $d + 1$ in which all the old edges between Poisson structures $P(\varrho, [a])$ head to arrowtail vertices for $\varrho$ but not to terminal vertices for the Casimirs $a_\ell$.

Indeed, it is the differential polynomials from these micro-graphs which, staying nonzero, retract to the bottom-most solution.

---

[14]The projection – from micro-graphs to differential-polynomial coefficients (in $\varrho$ and $a_\ell$) of multi-vectors – has a kernel which contains, as a strict subset, the space of Leibniz and zero (micro)graphs, but does not amount only to it. Can the dimension reduction result in an identically zero vector field $\vec{X}_{d-1}^\gamma([\varrho], [a]) \equiv \vec{0}$ on $\mathbb{R}^{d-1}$? That is, can a tower of micro-graph solutions $X_{d \geqslant d_0}^\gamma$ start at bottom dimension $d_0 > 2$ above the main case of $\mathbb{R}^2$?

*Remark* 7. Another constraint – upon the derivative order profiles in $X^i([\varrho],[a])$, hence in $[\![P,\vec{X}]\!]$ – comes from the bi-vector $Q_{\gamma_3}(P(\varrho,[a]))$ with known differential-polynomial coefficients. In effect, terms in $X^i$ can contain only those orders of derivatives which, under $[\![P,\cdot]\!]$, reproduce the actually existing profiles of derivatives in $Q_{\gamma_3}([\varrho],[a])$.

**Open problem 1.** Is there a dimension $d+1 < \infty$ at which the tetrahedral-graph flow on the space of Nambu structures over $\mathbb{R}^{d+1}$ becomes nontrivial in the second Poisson cohomology?

# Acknowledgements

The authors thank the organizers of GROUP34 colloquium on group theoretical methods in Physics on 18–22 July 2022 in Strasbourg for a very warm atmosphere during the meeting. The authors thank M. Kontsevich and G. Kuperberg for helpful discussions; the authors are grateful to the referee for useful comments and suggestions.

**Funding information**   A part of this research was done while A.K. was visiting at the IHÉS; A.K. thanks the IHÉS for hospitality, and thanks the IHÉS and Nokia Fund for financial support. The travel of A.K. was partially supported by project 135110 at the Bernoulli Institute, University of Groningen. The research of R.B. was supported by project 5020 at the Institute of Mathematics, Johannes Gutenberg–Universität Mainz and by CRC-326 grant GAUS "Geometry and Arithmetic of Uniformized Structures".

# A  (Non)triviality of $\gamma_3$-flow for Nambu–Poisson brackets on $\mathbb{R}^4$

From [4] we know that Kontsevich's tetrahedral $\gamma_3$-flow restricts to the space of Nambu-determinant Poisson bi-vectors $P(\varrho,[a_1],[a_2])$ over $\mathbb{R}^4$: the differential-polynomial velocities $\dot{\varrho}$ and $\dot{a}_1,\dot{a}_2$ inducing the graph cocycle evolution $\dot{P}([\varrho],[a_1],[a_2])$ are stored externally.[15] The evolutions $\dot{a}_1,\dot{a}_2(\varrho,[a_1],[a_2])$ are realized by Kontsevich graphs $\vec{\text{Or}}(\gamma_3)(P\otimes P\otimes P \otimes a_\ell)$, hence they are immediately expanded to micro-graph realizations. The evolution $\dot{\varrho}([\varrho], [a_1],[a_2])$ can then be expressed by using micro-graphs with minimal effort.

The problem of Poisson (non)triviality of the tetrahedral $\gamma_3$-flow for Nambu brackets in dimension $d = 4$ is open. At the level of micro-graphs and Nambu-determinant Poisson structures $P(\varrho,[a_1],[a_2])$ over $\mathbb{R}^4$, a solution $X^\gamma$ of (1) for the graph cocycle $\gamma_3$ would be realized by micro-graphs possibly with tadpoles, on one sink of in-degree 1, three vertices of out-degree 4, and two triples of terminal vertices for Casimirs $a_1,a_1,a_1$ and $a_2,a_2,a_2$.

**Proposition 8.** • There are $1,079$ isomorphism classes of directed graphs on one sink, three vertices of out-degree four, six terminal vertices, and at most one tadpole (of them, 352 are without a tadpole and 727 have one tadpole).
• Taking those graphs containing a vertex of in-degree one (for the sink), and dynamically appointing the Casimirs from the multi-set $\{a_1,a_1,a_1,a_2,a_2,a_2\}$ to the six terminal vertices of the above graphs, we obtain $38,120$ micro-graphs.
• Excluding repetitions in the above set of micro-graphs (e.g., if those micro-graphs are isomorphic), still not excluding micro-graphs which equal minus themselves under a symmetry (automorphism of micro-graph with outgoing edge ordering and known location of $a_1$'s and $a_2$'s) we obtain $19,957$ micro-graphs in the ansatz for $X^\gamma$ that would encode the trivializing vector field solutions, if any, of the coboundary equation $Q_{\gamma_3}\big(P(\varrho,[a_1],[a_2])\big) = [\![P,\vec{X}_4^{\gamma_3}([\varrho],[a_1],[a_2])]\!]$.

---

[15]https://rburing.nl/gcaops/adot_rhodot_g3_4D.txt

• Of these $19,957$ micro-graphs, one tadpole is present in $13,653$ micro-graphs, and there are no tadpoles in $6,304$ micro-graphs.

*Construction sketch.* The representatives of isomorphism classes of graphs without tadpoles are generated by the `nauty` command-line call `geng 10 9:12 | directg -e12 | pickg -d0 -m7 -D4 -M3` [7]. Likewise, the graphs with one tadpole are generated by first producing graphs with one edge fewer, using `geng 10 0:12 | directg -e11 | pickg -d0 -m7 -D4`, and adding a tadpole to the lonely vertex of out-degree 3 in each graph. For the appointment of Casimirs to 6 vertices in all *different* ways, one uses an efficient algorithm to generate the 20 permutations of the multi-set $\{a_1, a_1, a_1, a_2, a_2, a_2\}$. □

**Proposition 9** (R. Buring, PhD thesis (2022)). *If $d = 2$, the Poisson cocycles $Q_\gamma(P)$ for graph cocycles $\gamma \in \{\gamma_3, \gamma_5, \gamma_7\}$ are Poisson-trivial: $Q_\gamma(P) = [\![P, \vec{X}_2^\gamma(P)]\!]$. Every such vector field $\vec{X}_2^\gamma(P)$ is Hamiltonian w.r.t. the standard symplectic structure $\omega = \mathrm{d}x \wedge \mathrm{d}y$ on $\mathbb{R}^2$ and Hamiltonian $H^\gamma(P)$. The differential polynomials $H^\gamma(P)$ are encoded by sums of Kontsevich graphs.*

The case of $\gamma_3$ was known to Kontsevich [5], and the respective Hamiltonian was found by Bouisaghouane (see `arXiv:1702.06044` [math.DG]). The cases of $\gamma_5$ and chosen representative for the graph cocycle $\gamma_7$ are new.

**Example 5.** Let $P = u\partial_x \wedge \partial_y$ be the generic Poisson bi-vector on $\mathbb{R}^2$; we have $H^{\gamma_3} = 8u_y^2 u_{xx} - 16u_x u_y u_{xy} + 8u_x^2 u_{yy}$ and $H^{\gamma_5} = 6u_y^2 u_{xx} u_{xy}^2 - 12u_x u_y u_{xy}^3 - 6u_y^2 u_{xx}^2 u_{yy}$
$+ 12u_x u_y u_{xx} u_{xy} u_{yy} + 6u_x^2 u_{xy}^2 u_{yy} - 6u_x^2 u_{xx} u_{yy}^2 - 2u_y^3 u_{xy} u_{xxx} + 2u_x u_y^2 u_{yy} u_{xxx} + 2u_y^3 u_{xx} u_{xxy}$
$+ 2u_x u_y^2 u_{xy} u_{xxy} - 4u_x^2 u_y u_{yy} u_{xxy} - 4u_x u_y^2 u_{xx} u_{xyy} + 2u_x^2 u_y u_{xy} u_{xyy} + 2u_x^3 u_{yy} u_{xyy}$
$+ 2u_x^2 u_y u_{xx} u_{yyy} - 2u_x^3 u_{xy} u_{yyy} - 2u_y^4 u_{xxxx} + 8u_x u_y^3 u_{xxxy} - 12u_x^2 u_y^2 u_{xxyy} + 8u_x^3 u_y u_{xyyy}$
$- 2u_x^4 u_{yyyy}$.

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
