# Peer review of "The tower of Kontsevich deformations for Nambu-Poisson structures on $\mathbb{R}^{d}$: dimension-specific micro-graph calculus"

_SciPost Physics Proceedings, doi:SciPost Phys. Proc. 14, 020 (2023)_

## Round 2 · Referee Report · Kevin Morand · 2023-7-25

# REPORT ON THE SECOND VERSION OF "THE TOWER OF KONTSEVICH DEFORMATIONS FOR NAMBU-POISSON STRUCTURES ON $\mathbb{R}^d$: DIMENSION-SPECIFIC MICRO-GRAPH CALCULUS"

## 1. SUMMARY.

This updated version of the manuscript presents a more streamlined version of the obtained results. After recalling some motivating known results in $d = 2$ (Section 1), the bulk of the paper (Section 2) is devoted to a detailed proof (or rather hints of the proof, the core of the proof relying largely on brute-force computations) of Theorem 2 (first stated in [1]) stating the triviality of the tetrahedral flow on $d = 3$ Nambu-determinant Poisson bivectors by displaying an explicit trivialising vector. The proof relies on the use of "micro-graphs" (Definition 2) whose balancing yields a (large) system of algebraic equations to be solved. A simplifying assumption is made to reduce the computing burden, namely that the velocities of $\rho$ and the Casimir function $a$ obtained from the explicit expression of the tetrahedral Nambu flow are obtained from the evolution of $\rho$ and $a$ themselves with respect to the seeked-for trivialising vector. This assumption allows to find an explicit solution, given both in component expression and in terms of micro-graphs (*i.e.* in a manifestly GL(3)-invariant way). Suitable dimensional reduction is shown to reproduce the sunflower graph in two dimensions. It is then confirmed (Proposition 6) that the trivialisation occurring in $d = 3$ does not simply rely on universal (*i.e.* GL($\infty$)-invariant) mechanisms but rather on dimension-dependent ones. This prompts the formulation of a new conjecture (phrased as Open Problem 1) stating the existence of a dimensional threshold above which trivialisation would fail. Appendix A presents an attempt to check the conjecture for $d = 4$, however the computing complexity thereof makes it inconclusive. Nevertheless, it is hoped that the effort spent in the present work in describing the intricacies of the trivialisation mechanism in $d = 2, 3$ will in the future be rewarded by a successful attempt at unraveling an underlying mechanism susceptible of generalisations to higher dimensions.

## 2. OPINION.

The manuscript has been considerably rewritten and is overall better organised and easier to read than its previous version. The emphasis on marginally relevant topics (*e.g.* tadpoles and cardinalities of sets of graphs) has been reduced and the distinction between GL($\infty$) and GL($d$)-invariant methods clarified. In view of the above remarks, I recommend publication in SciPost Physics Proceedings.

## 3. SUGGESTIONS.

Here is a list of suggestions addressed to the authors, the latter remaining free to adopt these proposals at their own discretion.

(1) It would be worth to recall around Proposition 1 that:
  (a) For the same reason that any bivector is automatically Poisson in $d = 2$, any flow mediated by a Kontsevich graph of weight zero (but not necessarily a cocycle for Kontsevich's differential) will preserve the (trivial) Poisson condition.
  (b) Yet, the content of Proposition 1 is not trivial since the second order Lichnerowicz cohomology does not vanish *a priori* (*i.e.* $H^2(\pi) \neq 0$) [see *e.g.* [2]].

(2) An explicit component expression of the sunflower vector field would be useful, e.g. $\sigma^n = \partial_m(\partial_k P^{ij} \partial_i P^{kl} \partial_{jl} P^{mn})$.

(3) Proposition 3 would be more appropriately rephrased as a Remark.

(4) The Conclusion would be more appropriately rephrased as Concluding remarks.

(5) If length is considered an issue by the editors, both the proof of Proposition 1 and Appendix A can be removed (or at least considerably reduced) without altering the flow of the paper.

## 4. Non-mathematical points (typos, general comments)

(1) The expression "Civita symbol" used repeatedly throughout the text is slightly unusual, considering that T. Levi–Civita is a single individual.

(2) Bottom of p:5 *trivialzing* → *trivializing*

(3) The sum $10 = \cdots$ in Example 4 does not add up.

(4) Bottom of p:8 *Casimisrs* → *Casimirs*

## References

[1] R. Buring, D. Lipper, A. V. Kiselev "*The hidden symmetry of Kontsevich's graph flows on the spaces of Nambu-determinant Poisson brackets*" Open Communications in Nonlinear Mathematical Physics **2** 186-215 (2022) arXiv:2112.03897

[2] P. Monnier "*Poisson cohomology in dimension two*" arXiv:math/0005261

---

## Round 2 · Author Response

By following the Editorial Recommendation, the authors have revised
the submission. The authors are grateful to the Editor-in-Charge and
to the referee; comments of the referee were taken into account when
improving the text (now 10 pages as required), see the List of changes.
Applications of the apparent Poisson-triviality of the
graph-cocycle flows under study will be discussed in subsequent
publication(s), concerning in particular the deformation quantization
of Nambu--Poisson structures. The authors agree with referee's opinion
that it would be interesting to focus not only on the wheel cocycles:
e.g., the referee points out the Kontsevich--Shoikhet cocycle in this
context.

---

## Round 2 · List of Changes

In particular, section 1 is extended with Proposition 1 and its
explicit proof. As suggested by the referee, in section 2 we recall
the formula of trivializing vector field (now Theorem 2) to make this
article self-contained. In the encoding of this vector field by using
micro-graphs, sinks are properly indicated. The calculus of
micro-graphs is now introduced with greater care and in more detail.
The list of literature references is updated; technical Proposition
8 is moved to the last page. In new Proposition 9, the object of this
research is furthered to other wheel cocycles in the Kontsevich graph
complex.

Resubmission 2212.08063v3 on 16 August 2023

You are currently on this page

Resubmission 2212.08063v2 on 14 July 2023

---

## Round 3 · List of Changes

- All the typos which have been pointed out are fixed, and instances of "Civita symbol" are replaced with "Levi-Civita symbol".
- Proposition 1 is extended with a reference to the second Poisson cohomology and with an encoding and formula of the "sunflower" graph and its 1-vector field.

You are currently on this page

Resubmission 2212.08063v3 on 16 August 2023

---

## Editorial Decision

published